# Larger but younger fish when growth outpaces mortality in heated ecosystem

Max Lindmark[1]*[†], Malin Karlsson[1], Anna Gårdmark[2]

[1]Swedish University of Agricultural Sciences, Department of Aquatic Resources, Institute of Coastal Research, Öregrund, Sweden; [2]Swedish University of Agricultural Sciences, Department of Aquatic Resources, Uppsala, Sweden

**Abstract** Ectotherms are predicted to 'shrink' with global warming, in line with general growth models and the temperature-size rule (TSR), both predicting smaller adult sizes with warming. However, they also predict faster juvenile growth rates and thus larger size-at-age of young organisms. Hence, the result of warming on the size-structure of a population depends on the interplay between how mortality rate, juvenile- and adult growth rates are affected by warming. Here, we use two-decade long time series of biological samples from a unique enclosed bay heated by cooling water from a nearby nuclear power plant to become 5–10 °C warmer than its reference area. We used growth-increment biochronologies (12,658 reconstructed length-at-age estimates from 2426 individuals) to quantify how >20 years of warming has affected body growth, size-at-age, and catch to quantify mortality rates and population size- and age structure of Eurasian perch (*Perca fluviatilis*). In the heated area, growth rates were faster for all sizes, and hence size-at-age was larger for all ages, compared to the reference area. While mortality rates were also higher (lowering mean age by 0.4 years), the faster growth rates lead to a 2 cm larger mean size in the heated area. Differences in the size-spectrum exponent (describing how the abundance declines with size) were less clear statistically. Our analyses reveal that mortality, in addition to plastic growth and size-responses, is a key factor determining the size structure of populations exposed to warming. Understanding the mechanisms by which warming affects the size- and the age structure of populations is critical for predicting the impacts of climate change on ecological functions, interactions, and dynamics.

*For correspondence:
max.lindmark@slu.se

Present address: [†]Max Lindmark, Swedish University of Agricultural Sciences, Department of Aquatic Resources, Institute of Marine Research, Lysekil, Sweden

Competing interest: The authors declare that no competing interests exist.

## Editor's evaluation

This work provides convincing evidence to refute a general tenet in biology, that warming induces smaller maximum body sizes in adult ectoterm individuals. Using a semi-natural experiment in an exceptional man-made ecosystem, the authors demostrate that fish in waters warmed by a nearby nuclear plant grew faster but died younger, causing little effect on the size distribution of fish in the area. This work will be of interest to ecologists and physiologists interested in the impacts of global warming on natural communities.

## Introduction

Ectotherm species, constituting 99% of species globally (*Atkinson and Sibly, 1997*; *Wilson, 1992*), are commonly predicted to shrink in a warming world (*Gardner et al., 2011*; *Sheridan and Bickford, 2011*). However, as the size distribution of many species spans several orders of magnitude, and temperature effects on size may depend on size or age, it is important to be specific about which sizes- or life stages are predicted to shrink (usually mean or adult is meant). For instance, warming can shift size distributions without altering mean size if increases in juvenile size-at-age outweigh the decline in size-at-age in adults, which is consistent with the temperature size rule, TSR (*Atkinson,*

1994). Resolving how warming induces changes in population's size distribution may thus be more instructive (*Fritschie and Olden, 2016*), especially for inferring warming effects on species' ecological role, biomass production, or energy fluxes (*Gårdmark and Huss, 2020*; *Yvon-durocher et al., 2011*). This is because key processes such as metabolism, feeding, growth, and mortality scale with body size (*Andersen and Link, 2020*; *Blanchard et al., 2017*; *Brown et al., 2004*; *Pauly, 1980*; *Thorson et al., 2017*; *Ursin, 1967*). Hence, as the value of these traits at mean body size is not the same as the mean population trait value (*Bernhardt et al., 2018*), the size distribution within a population matters for its dynamics and for how it changes under warming.

The population size distribution can be represented as a size-spectrum, which generally is the frequency distribution of individual body sizes (*Edwards et al., 2017*). It is often described in terms of the size-spectrum slope (slope of individuals or biomass of a size class over the mean size of that class on a log-log scale [*Edwards et al., 2017*; *Sheldon et al., 1973*; *White et al., 2007*]) or simply the exponent of the power law individual size distribution (*Edwards et al., 2017*). The size-spectrum thus results from temperature-dependent ecological processes such as body growth, mortality, and recruitment (*Blanchard et al., 2017*; *Heneghan et al., 2019*). Despite its rich theoretical foundation (*Andersen, 2019*) and usefulness as an ecological indicator (*Blanchard et al., 2005*), few studies have evaluated warming effects on the species size-spectrum in larger-bodied species (but see *Blanchard et al., 2005*), and none in large scale experimental set-ups. There are numerous paths by which a species' size-spectrum could change with warming (*Heneghan et al., 2019*). For instance, in line with TSR predictions, warming may lead to a smaller size-spectrum exponents (steeper slope) if the maximum size declines. However, changes in size-at-age and the relative abundances of juveniles and adults may alter this decline in the size-spectrum slope. Warming can also lead to elevated mortality (*Barnett et al., 2020*; *Berggren et al., 2022*; *Biro et al., 2007*; *Pauly, 1980*), partly because a faster pace of life with higher metabolic rates is associated with a shorter lifespan (*Brown et al., 2004*; *Munch and Salinas, 2009*) or due to direct lethal effects of extreme temperature events. This truncates the age distribution towards younger individuals (*Barnett et al., 2017*), which may reduce density dependence and potentially increase growth rates, thus countering the effects of mortality on the size-spectrum exponent. However, not all sizes may benefit from warming, as e.g. the optimum temperature for growth declines with size (*Lindmark et al., 2022*). Hence, the effect of warming on the size-spectrum depends on several interlinked processes affecting abundance-at-size and size-at-age.

Size-at-age is generally predicted to increase with warming for small individuals, but decrease for large individuals according to the mentioned TSR (*Atkinson, 1994*; *Ohlberger, 2013*). Several factors likely contribute to this pattern, such as increased allocation to reproduction (*Wootton et al., 2022*) and larger individuals in fish populations having optimum growth rates at lower temperatures (*Lindmark et al., 2022*). Empirical support in fishes for this pattern seems to be more consistent for increases in size-at-age of juveniles (*Huss et al., 2019*; *Rindorf et al., 2008*; *Thresher et al., 2007*) than declines in adult size-at-age (but see *Baudron et al., 2014*; *Oke et al., 2022*; *Smoliński et al., 2020*), for which a larger diversity in responses is observed among species (*Barneche et al., 2019*; e.g., *Huss et al., 2019*). However, most studies have been done on commercially exploited species, since long-time series are more common in such species. This may confound or interact with the effects of temperature because fishing mortality can affect density-dependent growth (*van Gemert et al., 2018*), but also select for slow-growing individuals and changes in maturation processes, which also influences growth trajectories (*Audzijonyte et al., 2016*).

The effect of temperature on mortality rates of wild populations is often studied using among-species analyses (*Pauly, 1980*; *Thorson et al., 2017*). These relationships based on thermal gradients in space may not necessarily be the same as the effects of *warming* on mortality in single populations. Hence, the effects of warming on growth and size-at-age, and mortality within natural populations constitute a key knowledge gap for predicting the consequences of climate change on population size spectra.

Here, we used data from a unique, large-scale 23-year-long heating experiment of a coastal ecosystem to quantify how warming changed fish body growth, mortality, and the size structure in an unexploited population of Eurasian perch (*Perca fluviatilis*, 'perch'). We compare fish from this enclosed bay exposed to temperatures approximately 5–10 °C above normal ('heated area') with fish from a reference area in the adjacent archipelago (*Figure 1*). Using hierarchical Bayesian models, we

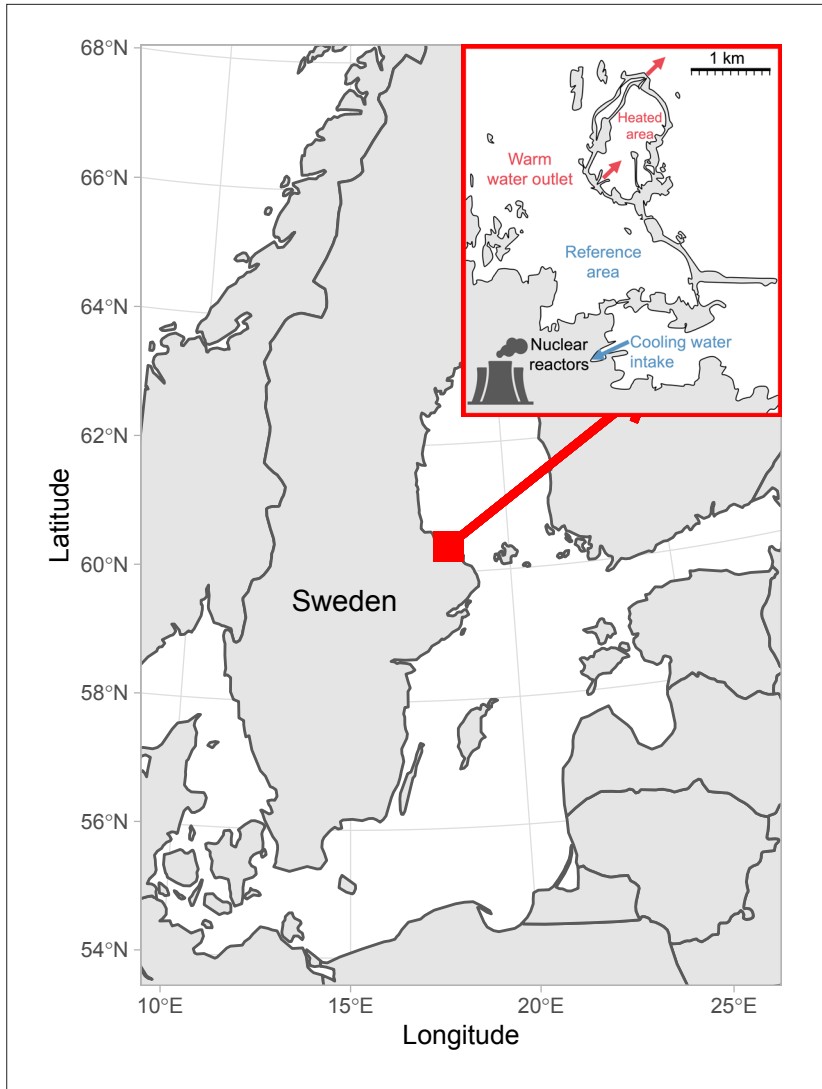

**Figure 1.** Map of the area with the unique whole-ecosystem warming experiment from which perch in this study was sampled. Inset shows the 1 km² enclosed coastal bay that has been artificially heated for 23 years, the adjacent reference area with natural temperatures, and locations of the cooling water intake, and where the heated water outlet from nuclear power plants enters the heated coastal basin. The arrows indicate the direction of water flow.

quantify differences in key individual- and population-level parameters, such as body growth, asymptotic size, mortality rates, and size spectra, between the heated and reference coastal areas.

## Results

Analysis of perch size-at-age using the von Bertalanffy growth equation (VBGE) revealed that fish cohorts (year classes) in the heated area both grew faster initially (larger size-at-age) and reached larger predicted asymptotic sizes than those in the unheated reference area (*Figure 2*). The model with area-specific VBGE parameters ($L_\infty$ , $K$, and $t_0$) had the best out-of-sample predictive accuracy (the largest expected log pointwise predictive density for a new observation; *Supplementary file 1a*). Models where both $L_\infty$ and $K$ were shared did not converge (*Supplementary file 1a*). Both the estimated values for fish asymptotic length ($L_\infty$) and growth coefficient ($K$) were larger in the heated compared to the reference *area* (*Figure 2—figure supplement 8*). *We estimated that* the asymptotic length of fish in the heated area was 16% larger than in the reference area (calculated as $\frac{L_{\infty\text{heat}} - L_{\infty\text{ref}}}{L_{\infty\text{ref}}}$) ($L_{\infty\text{heat}} = 45.7 \left[36.8, 56.3\right]$, $L_{\infty\text{ref}} = 39.4 \left[35.4, 43.9\right]$ , where the point estimate is the posterior median and values in brackets correspond to the 95% credible interval). The growth coefficient was 27%

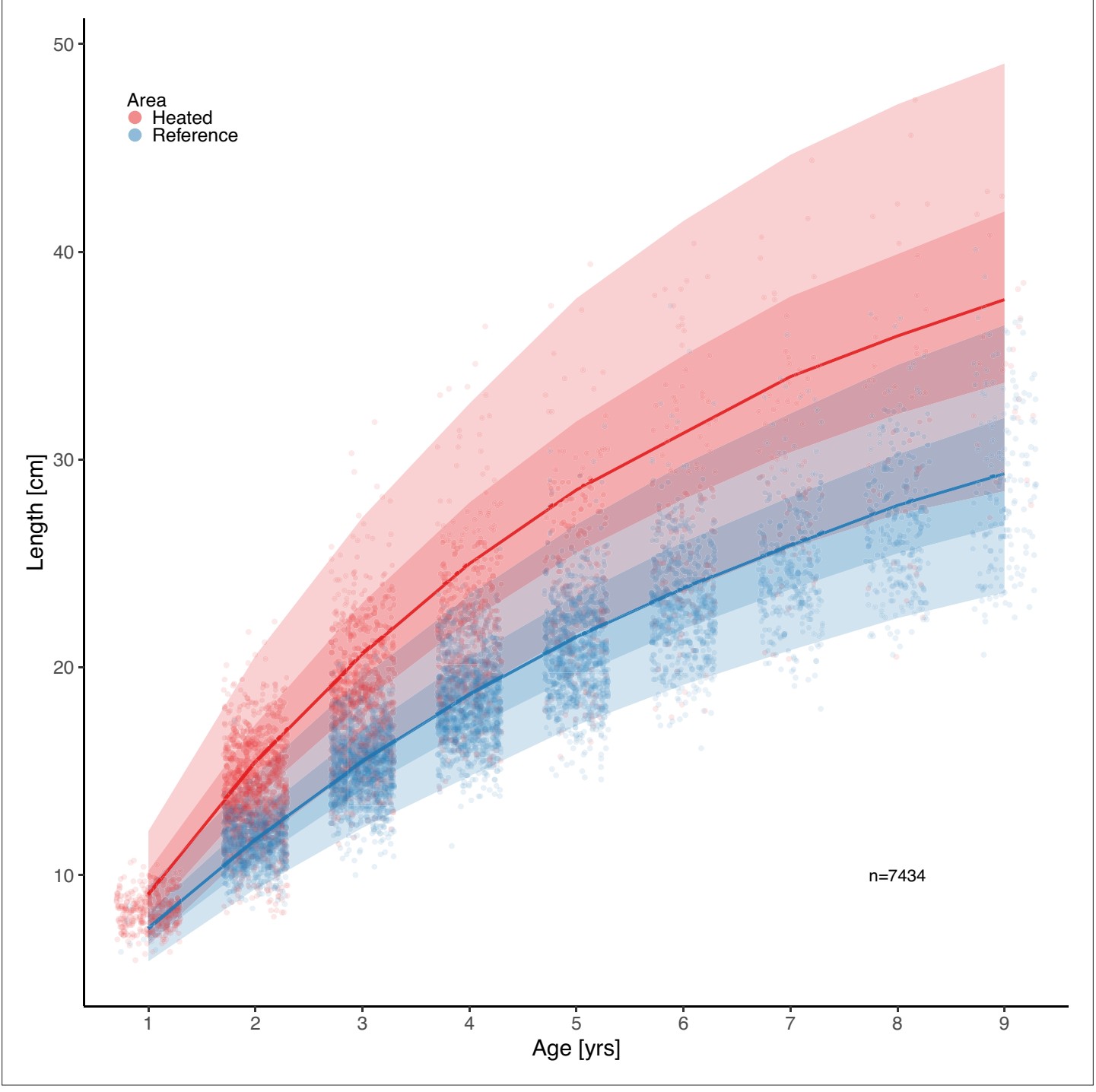

**Figure 2.** Fish grow faster and reach larger sizes in the heated enclosed bay (red) compared to the reference area (blue). Points depict individual-level length-at-age and lines show the median of the posterior draws of the global posterior predictive distribution (without group-level effects), both exponentiated, from the von Bertalanffy growth model with area-specific coefficients. The shaded areas correspond to 50% and 90% credible intervals.

The online version of this article includes the following figure supplement(s) for figure 2:

**Figure supplement 1.** Prior predictive distribution for the von Bertalanffy growth equation (posterior draws from the prior only, ignoring the likelihood).

**Figure supplement 2.** The best model of the von Bertalanffy growth equation: (**A**) trace plot to illustrate chain convergence for key (population-level) parameters, (**B**) residuals, (**C**) QQ-plot, and (**D**) posterior predictive check.

**Figure supplement 3.** Cohort-specific predictions from the best von Bertalanffy model (i.e. with cohort-specific $L_\infty$ and $K$).

*Figure 2 continued on next page*

*Figure 2 continued*

**Figure supplement 4.** The average length-at-age is larger for fish of all ages in the heated enclosed bay compared to the reference area, and the relative difference declines very slightly with age.

**Figure supplement 5.** Posterior distributions of the cohort-varying $L_\infty$ parameter in the best von Bertalanffy growth model.

**Figure supplement 6.** Posterior distributions of the cohort-varying $K$ parameter in the von Bertalanffy model.

**Figure supplement 7.** Prior vs posterior distributions for parameters $L_\infty$ (**A**), $K$ (**B**) and $t_0$ (**C**) in the best model of the von Bertalanffy growth equation.

**Figure supplement 8.** Posterior distributions of $K$ (**A**) and $L_\infty$ (**B**) for both areas and the distribution of their difference (**C, D**).

**Figure supplement 9.** Analysis of sensitivity of including the most recent cohorts, with a smaller age range and, therefore, less certain estimates of $L_\infty$.

**Figure supplement 10.** Analysis of sensitivity of including the most recent cohorts, with a smaller age range and, therefore, less certain estimates of $L_\infty$.

larger in the heated area ($K_{heat} = 0.19[0.15, 0.23]$, $K_{ref} = 0.15[0.12, 0.17]$). These differences in growth parameters lead to fish being approximately 7%–11% larger in the heated area at any age relative to the reference area (*Figure 2—figure supplement 4*). Due to the last three cohorts (1995–1997) having large estimates of $L_{\infty heat}$ and low $K$ (potentially due to their negative correlation and because of the young age with data far from the asymptote, *Figure 2—figure supplements 3 and 5–6*), we fit the same model with these cohorts omitted to evaluate the importance of those for the predicted difference between the areas. Without these, the predicted difference in size-at-age was still clear, but smaller (between 4%–7%, *Figure 2—figure supplements 9–10*).

In addition, we found that growth rates in the reference area were both slower and declined faster with size compared to the heated area (*Figure 3*). The best model for growth ($G = \alpha L^\theta$) had area-specific $\alpha$ and $\theta$ parameters (*Supplementary file 1b*). Initial growth ($\alpha$) was estimated to be 18% faster in the heated than in the reference area ($\alpha_{heat} = 512\,[462, 565]$, $\alpha_{ref} = 433\,[413, 454]$), and the growth of fish in the heated area declines more slowly with length than in the reference area ($\theta_{heat} = -1.13\,[-1.16, -1.11]$, $\theta_{ref} = -1.18\,[-1.19, -1.16]$). The distribution of differences of the posterior samples for $\alpha$ and only had 0.3% and 0.2% of the density below 0, respectively (*Figure 3C and E*), indicating a high probability that length-based growth rates are faster in the heated area.

By analyzing the decline in catch-per-unit-effort over age, we found that the instantaneous mortality rate $Z$ (the rate at which log abundance declines with age) is higher in the heated area (*Figure 4*). $Z$ was estimated as a fixed effect, as the model where only intercepts varied among years had the best out-of-sample predictive ability. The overlap with zero is 0.07% for the distribution of differences between posterior samples of $Z_{heat}$ and $Z_{ref}$ (*Figure 4C*). We estimated $Z_{heat}$ to be 0.73 [0.66,0.79] and $Z_{ref}$ to be 0.62 [0.58,0.67], which corresponds to annual mortality rates (calculated as $1 - e^{-Z}$) of 52% in the heated area and 46% in the reference area.

Lastly, analysis of the size- and age-structure in the two areas revealed that, despite the faster growth rates, higher mortality, and larger maximum sizes in the heated area *Figure 5A*, the size-spectrum exponents were largely similar *Figure 5B, C*. In fact, the size-spectrum exponent was only slightly larger in the heated area (*Figure 5B*), and their 95% confidence intervals largely overlap. However, results from the lognormal model fitted to the size- and age-distributions revealed that the average size was two centimeters longer and the average age 0.4 years younger in the heated compared to the reference area (*Figure 6*).

## Discussion

Our study provides strong evidence for warming-induced differentiation in growth and mortality in a natural population of an unexploited, temperate fish species exposed to an ecosystem-scale experiment with 5–10°C above normal temperatures for more than two decades. Interestingly, these effects largely, but not completely, counteract each other when it comes to population size-structure—while the fish are younger, they are also larger on average. However, differences in the rate of decline in abundance with size are less pronounced between the areas. It is difficult to generalize these findings since it is a study of only a single species. It is, however, a unique climate change experiment, as experimental studies on fish to date are much shorter and often on scales much smaller than whole ecosystems, and long-time series of biological samples exist mainly for commercially exploited fish species (*Baudron et al., 2014*; *Smoliński et al., 2020*; *Thresher et al., 2007*) (in which fisheries

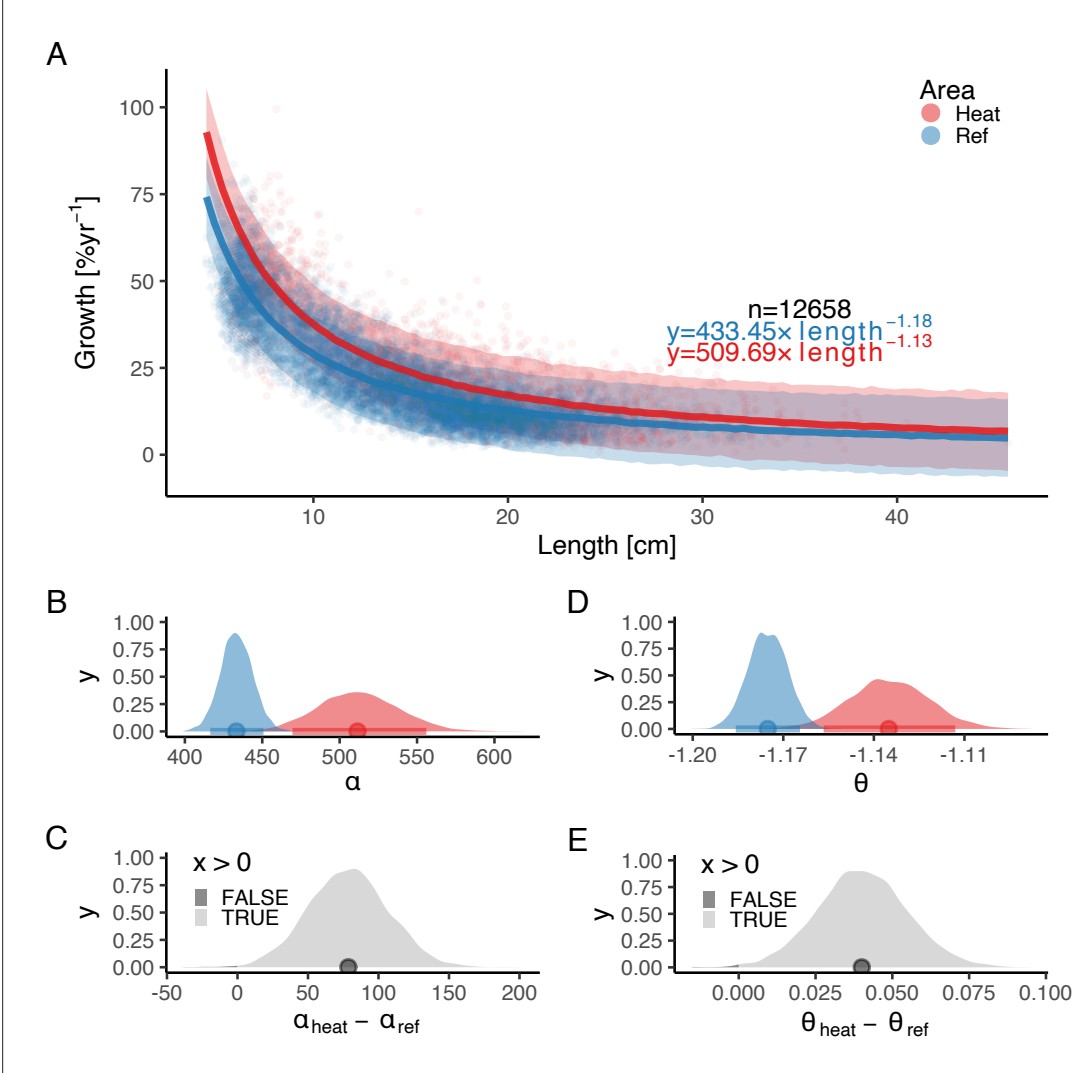

**Figure 3.** The faster growth rates in the heated area (red) compared to the reference (blue) are maintained as fish grow in size. The points illustrate specific growth rate estimated from back-calculated length-at-age (within individuals) as a function of length, expressed as the geometric mean of the length at the start and end of the time interval. Lines show the median of the posterior draws of the global posterior predictive distribution (without group-level effects) from the allometric growth model with area-specific coefficients. The shaded areas correspond to the 90% credible interval. The equation uses mean parameter estimates. Panel (**B**) shows the posterior distributions for initial growth ($\alpha_{heat}$ (red) and $\alpha_{ref}$ (blue)), and (**C**) the distribution of their difference. Panel (**D**) shows the posterior distributions for the allometric exponent ($\theta_{heat}$ and $\theta_{ref}$), and (**E**) the distribution of their difference. The fill color depicts the area below 0 (0.3% and 0.2% for $\alpha$ and $\theta$, respectively).

The online version of this article includes the following figure supplement(s) for figure 3:

**Figure supplement 1.** Prior predictive distribution for the allometric growth model (posterior draws from the prior only, ignoring the likelihood).

**Figure supplement 2.** The best allometric growth model: (**A**) trace plot to illustrate chain convergence for key (population-level) parameters, (**B**) residuals, (**C**) QQ-plot, and (**D**) posterior predictive check.

**Figure supplement 3.** Prior vs posterior distributions for parameters $\alpha$ (**A**) and $\theta$ (**B**) in the best allometric growth model (inset in panel (**B**) is a zoomed-in version to better visualize the priors in the range of the posteriors).

exploitation affects size-structure both directly and indirectly by selecting for fast-growing individuals). While factors other than temperature could have contributed to the observed elevated growth and mortality, the temperature contrast is unusually large for natural systems (i.e. 5–10°C, which can be compared to the 1.35°C change in the Baltic Sea between 1982 and 2006 [*Belkin, 2009*]). Moreover, heating occurred at the scale of a whole ecosystem, which makes the findings highly relevant in the context of global warming.

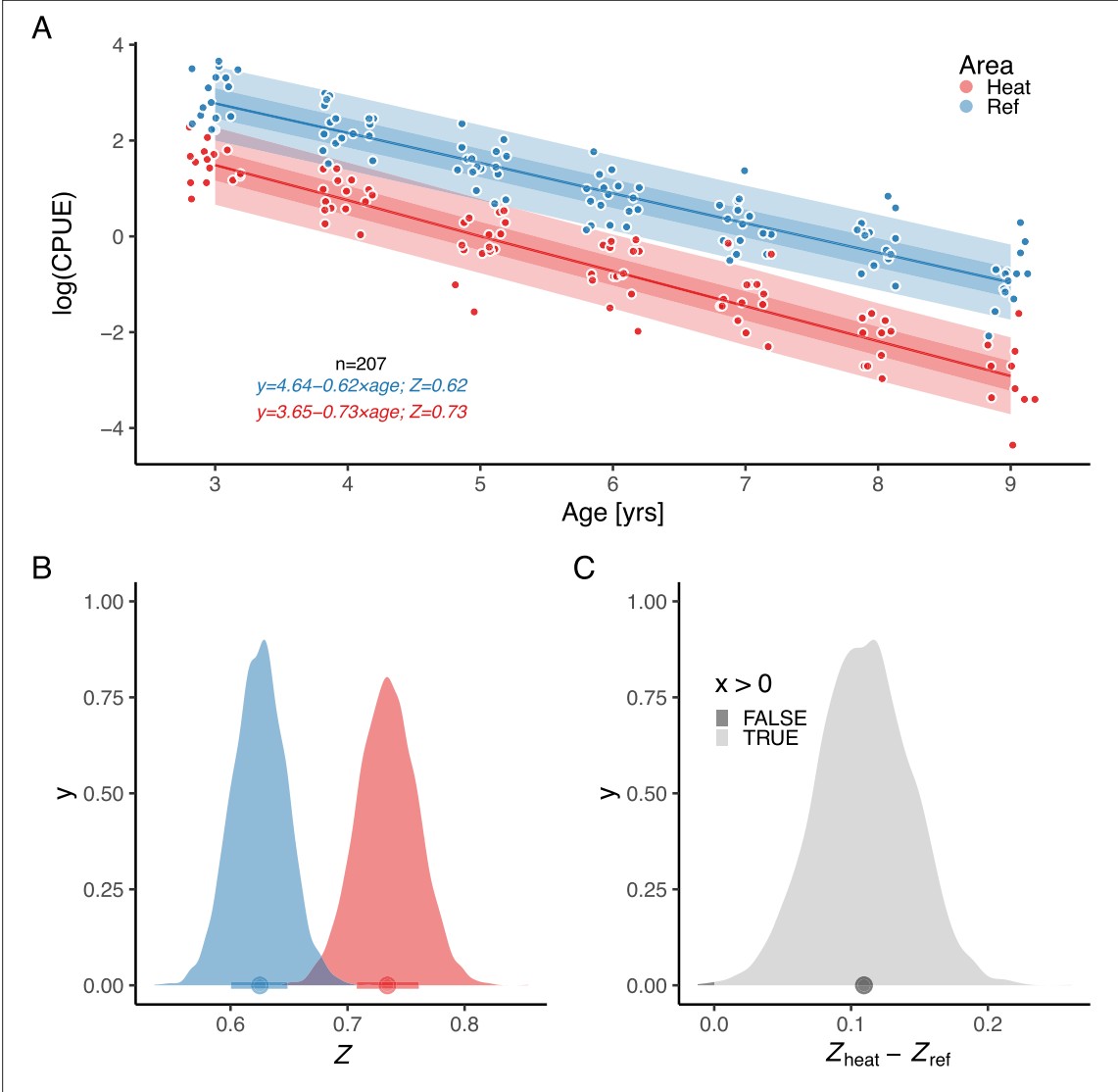

**Figure 4.** The instantaneous mortality rate ($Z$) is higher in the heated area (red) than in the reference (blue). Panel (**A**) shows $\log\left(CPUE\right)$ as a function of *age*, where the slope corresponds to $-Z$. Lines show the median of the posterior draws of the global posterior predictive distribution (without group-level effects) and the shaded areas correspond to the 50% and 90% credible intervals. The equation shows mean parameter estimates. Panel (**B**) shows the posterior distributions for mortality rate ($Z_{heat}$ and $Z_{ref}$), and (**C**) the distribution of their difference, where the fill color depicts the area below 0 (0.07%).

The online version of this article includes the following figure supplement(s) for figure 4:

**Figure supplement 1.** Catch per unit effort (CPUE) as a function of age, by area and year, for determining which ages are representatively caught by the fishing gear.

**Figure supplement 2.** The best catch curve model: (**A**) trace plot to illustrate chain convergence for key (population-level) parameters, (**B**) residuals, (**C**) QQ-plot, and (**D**) posterior predictive check.

Interestingly, our findings contrast with both broader predictions about declining mean or adult body sizes based on the GOLT hypothesis (*Cheung et al., 2013*; *Pauly, 2021*), and with intraspecific patterns such as the TSR (temperature-size rule, *Atkinson, 1994*). The contrasts lie in that both asymptotic size and size-at-age of mature individuals, as well as the proportion of larger individuals, were slightly larger and higher in the heated area—despite the elevated mortality rates. This result was unexpected for two reasons: optimum growth temperatures generally decline with body size within species under food satiation in experimental studies (*Lindmark et al., 2022*), and fish tend to mature at smaller body sizes and allocate more energy into reproduction as it gets warmer (*Niu et al.,*

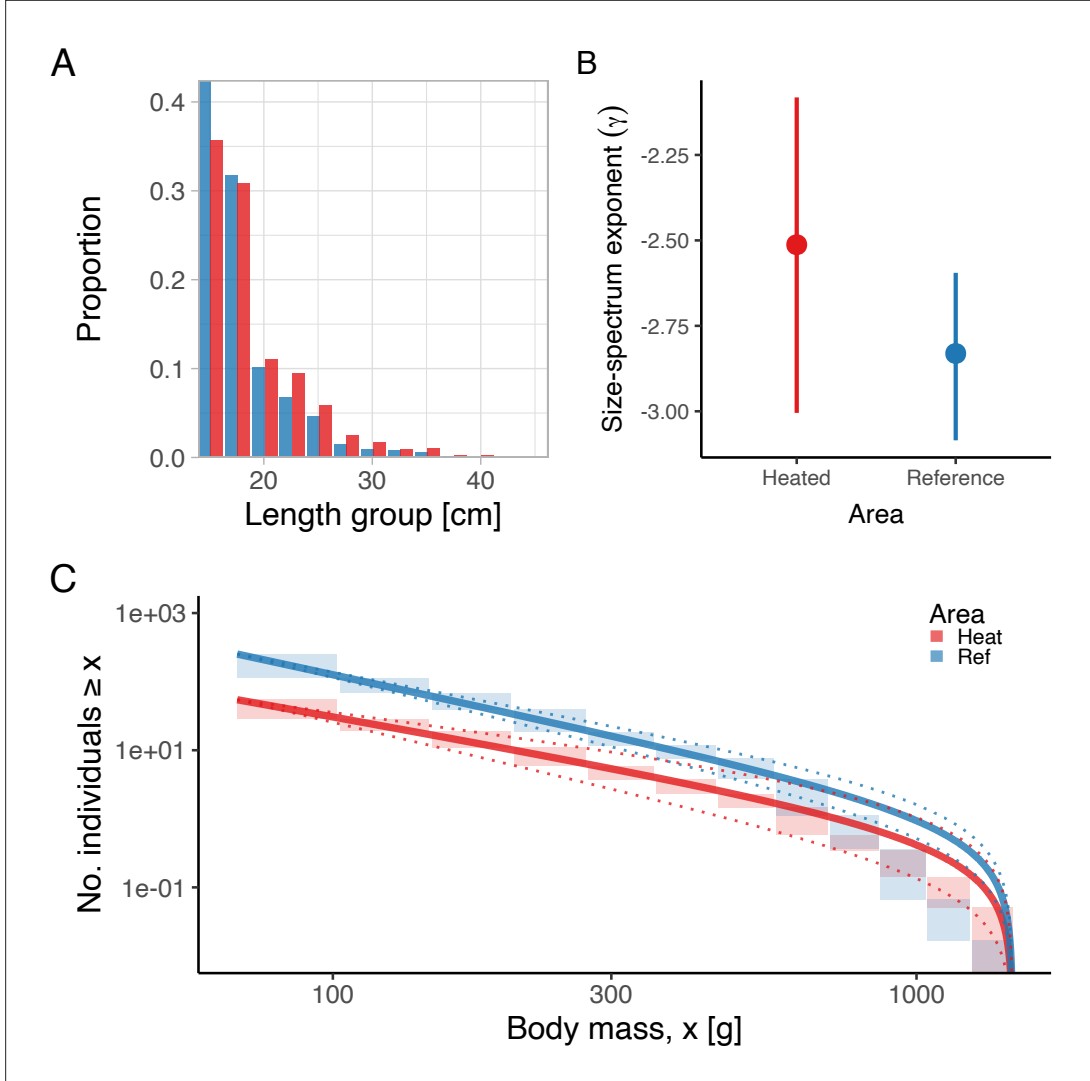

**Figure 5.** The heated area (red) has a larger proportion of large fish than the reference area (blue), illustrated both in terms of histograms of proportions at size (**A**) and the biomass size-spectrum (**B, C**), but the difference in the slope of the size spectra between the areas is not statistically clear (**C**). Panel (**A**) illustrates histograms of length groups in the heated and reference area as proportions (for all years pooled). Panel (**B**) shows the estimate of the size-spectrum exponent, $\gamma$, where vertical lines depict the 95% confidence interval. Panel (**C**) shows the size distribution and MLEbins fit (red and blue solid curves for the heated and reference area, respectively) with 95% confidence intervals indicated by dotted lines. The vertical span of rectangles illustrates the possible range of the number of individuals with body mass ≥ the body mass of individuals in that bin.

*2023*; *Wootton et al., 2022*). Both patterns have been used to explain how growth can increase for small and young fish, while large and old fish typically do not benefit from warming. Our study species is no exception to these rules (*Huss et al., 2019*; *Karås and Thoresson, 1992*; *Niu et al., 2023*; *Sandstrom et al., 1995*). This suggests that growth dynamics under food satiation may not be directly proportional to those under natural feeding conditions (*Railsback, 2022*). It could also mean that while temperatures is near optimum for growth in the warmest months of the year for a 15 cm individual (and above optimum for larger fish as the optimum declines with size) (*Huss et al., 2019*; *Lindmark et al., 2022*), the exposure to such high temperatures is not enough to cause strong reductions in growth and eventually size-at-age. Our results highlight that we need to focus on understanding to what extent the commonly observed increase in size-at-age for juveniles in warm environments can be maintained as they grow older.

Our finding that mortality rates were higher in the heated area was expected—warming leads to faster metabolic rates (faster 'pace of life'), which in turn is associated with a shorter life span (*Brown et al., 2004*; *McCoy and Gillooly, 2008*; *Munch and Salinas, 2009*). Extreme temperatures,

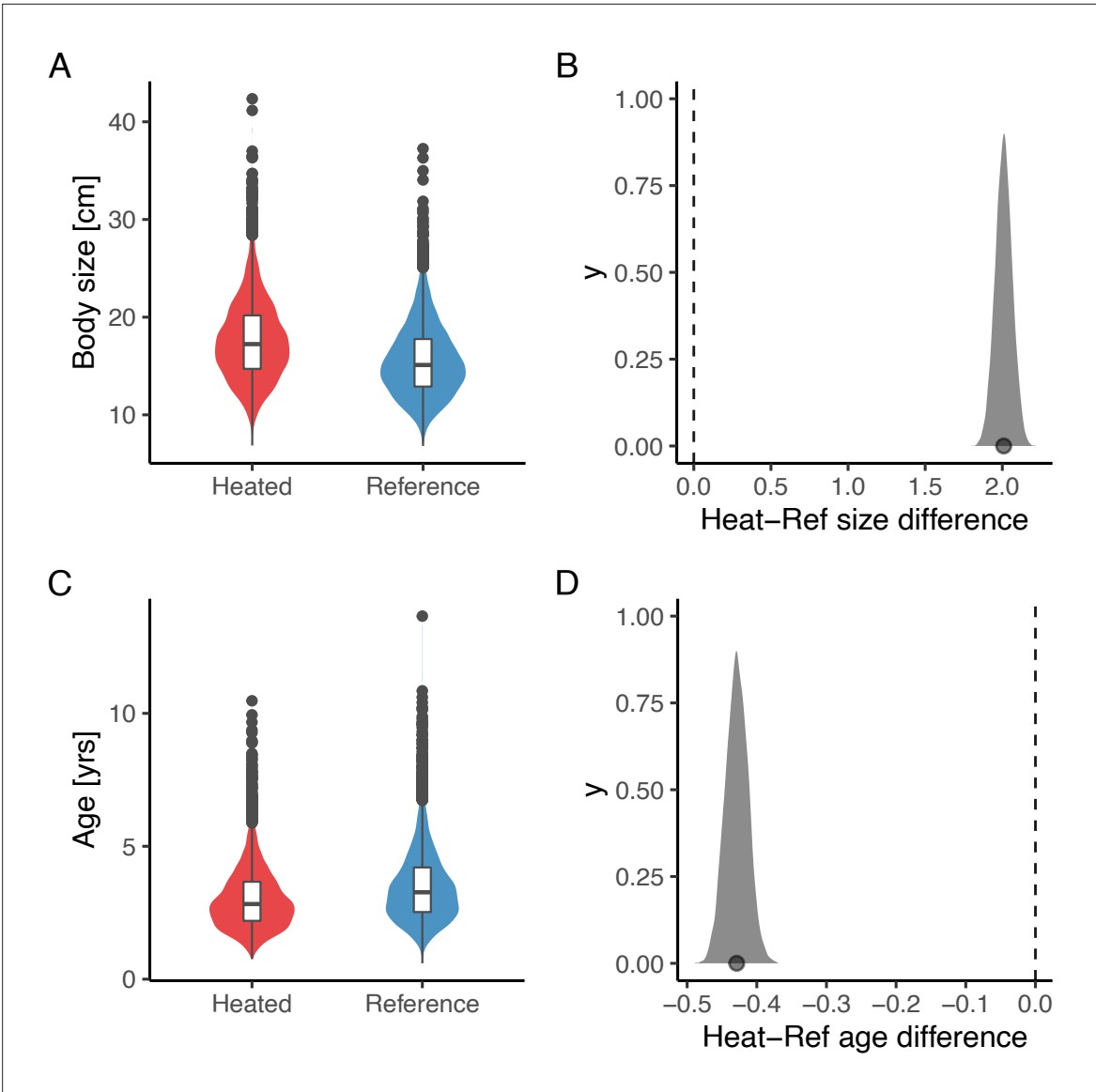

**Figure 6.** The average size is larger (**A, B**), but the average age (**C, D**) is younger in the heated area compared to the reference area. The violin plots (**A, C**) are based on draws from the global posterior predictive distribution (without group-level effects) for mean size and age from the lognormal model, respectively, with the random year effect omitted, while the density plots (**B, D**) depict the difference between areas based on draws from the expected value of the posterior predictive distribution. Hence, the latter has a smaller variation and the difference in means is more pronounced. The average size is 2 cm larger in the heated area, and the average age is 0.4 years younger (**B, D**).

The online version of this article includes the following figure supplement(s) for figure 6:

**Figure supplement 1.** Size (top) and age (bottom) distribution of catches, all years pooled, as used in the lognormal model estimate the mean size and catch.

**Figure supplement 2.** Lognormal length and age models model diagnostics and fit.

which may be more common in warmed systems under natural variability, can also be lethal if e.g., acute oxygen demands cannot be met (*Sandblom et al., 2016*). Warming may further increase predation mortality, as predators' feeding rates increase in order to meet the higher demand for food (*Biro et al., 2007*; *Pauly, 1980*; *Ursin, 1967*). However, most evidence to date of the temperature dependence of mortality rates in natural populations stems from across-species studies (*Gislason et al., 2010*; *Pauly, 1980*; *Thorson et al., 2017*, but see *Berggren et al., 2022*; *Biro et al., 2007*). Across-species relationships are not necessarily determined by the same processes as within-species

relationships; thus, our finding of warming-induced mortality in a heated vs control environment in two nearby con-specific populations is important.

Since a key question for understanding the implications of warming on ectotherm populations is if larger individuals in a population become rarer or smaller (*Ohlberger, 2013*; *Ohlberger et al., 2018*), within-species mortality and growth responses to warming need to be further studied. Importantly, this requires accounting also for the effects of warming on growth, and how responses in growth and mortality depend on each other. For instance, higher mortality (predation or natural, physiological mortality) can release intra-specific competition and thus increase growth. While e.g., benthic invertebrate density was not affected by the initial warming of the heated area (*Sandstrom et al., 1995*), warming-induced mortality may have led to higher benthic prey availability *per capita* for the studied perch. Conversely, altered growth and body sizes can lead to changes in size-specific mortality, such as predation or starvation, both of which are expected to change with warming (*Thunell, 2023*). In conclusion, individual-level patterns such as the TSR can only be used to predict changes in the population-level size structure in limited cases, as it does not concern changes in abundance-at-size via mortality. Mortality may, however, be an important driver of the observed shrinking of ectotherms (*Peralta-Maraver and Rezende, 2021*). Understanding the mechanisms by which the size- and age-distribution change with warming is critical for predicting how warming changes species functions and ecological roles (*Audzijonyte et al., 2020*; *Fritschie and Olden, 2016*; *Gårdmark and Huss, 2020*). Our findings demonstrate that a key to do this is to acknowledge temperature effects on both growth and mortality and how they interact.

## Materials and methods
### Data
We use size-at-age data from perch sampled annually from an artificially heated enclosed bay ('the Biotest basin') and its reference area, both in the western Baltic Sea (*Figure 1*). Heating started in 1980, the first analyzed cohort is 1981, and the first and last catch year is 1987 and 2003, respectively, to omit transient dynamics and acute responses, and to ensure we use cohorts that only experienced one of the thermal environments during its life. A grid at the outlet of the heated area (*Figure 1*) prevented fish larger than 10 cm from migrating between the areas (*Adill et al., 2013*; *Huss et al., 2019*), and genetic studies confirm the reproductive isolation between the two populations during this time period (*Björklund et al., 2015*). However, the grid was removed in 2004. Since then, fish growing up in the heated Biotest basin can easily swim out and fish caught in the reference area can no longer be assumed to be born there. Hence, we use data only up until 2003. This resulted in 12,658 length-at-age measurements from 2,426 individuals (i.e. multiple measurements per individual) from 256 net deployments.

We use data from fishing events using survey-gillnets that took place in October in the heated Biotest basin and in August in the reference area when temperatures are most comparable between the two areas (*Huss et al., 2019*), because temperature affects catchability in static gears. The catch was recorded by 2.5 cm length classes during 1987–2000, and into 1 cm length groups in years 2001–2003. To express lengths in a common length standard, 1 cm intervals were converted into 2.5 cm intervals. The unit of catch data is hence the number of fish caught by 2.5 cm size class per net per night (i.e. a catch-per-unit-effort (CPUE) variable). All data from fishing events with disturbance affecting the catch (e.g. seal damage, strong algal growth on the gears, clogging by drifting algae) were removed (years 1996 and 1999 from the heated area in the catch data).

Length-at-age throughout an individuals' life was reconstructed for a random or length-stratified subset of caught individuals each year (depending on which year, and in some cases, the number of fish caught). This was done using growth-increment biochronologies derived from annuli rings on the operculum bones, with control counts done on otoliths. Such analyses have become increasingly used to analyze changes in the growth and size-at-age of fishes (*Essington et al., 2022*; *Morrongiello and Thresher, 2015*). Specifically, an established power-law relationship between the distance of annual rings and fish length was used: $L = \kappa R^s$, where $L$ is the length of the fish, $R$ the operculum radius, $\kappa$ the intercept, and $s$ the slope of the line for the regression of log-fish length on log-operculum radius from a large reference data set for perch (*Thoresson, 1996*). Back-calculated length-at-age was obtained from the relationship $L_a = L_s \left(\frac{r_a}{R}\right)^s$, where $L_a$ is the back-calculated body length at age $a$, $L_s$ is the final

body length (body length at catch), $r_a$ is the distance from the center to the annual ring corresponding to age $a$ and $s = 0.861$ for perch (**Thoresson, 1996**). Since perch exhibits sexual size-dimorphism, and age determination together with back-calculation of growth was not done for males in all years, we only used females for our analyses.

## Statistical analysis

The differences in size-at-age, growth, mortality, and size structure between perch in the heated and the reference area were quantified using hierarchical linear and non-linear models fitted in a Bayesian framework. First, we describe each statistical model and then provide details of model fitting, model diagnostics, and comparison.

To describe individual growth throughout life, we fit the von Bertalanffy growth equation (VBGE) (**Beverton and Holt, 1957**; **von Bertalanffy, 1938**) on a log scale, describing length as a function of age to evaluate differences in size-at-age and asymptotic size: $\log(L_t) = \log\left(L_\infty\left(1 - e^{\left(-K(t-t_0)\right)}\right)\right)$, where $L_t$ is the length-at-age $t$ (years), $L_\infty$ is the asymptotic size, $K$ is the Brody growth coefficient ($yr^{-1}$) and $t_0$ is the age when the average length was zero. Here and henceforth, log refers to natural logarithms. We used only age and size at catch, i.e. not back-calculated length-at-age. This was to have a simpler model and not have to account for parameters varying within individuals as well as cohorts, as mean sample size per individual was only ~5. We let parameters vary among cohorts rather than year of catch, because individuals within cohorts share similar environmental conditions and density dependence (**Morrongiello and Thresher, 2015**). Eight models in total were fitted with area dummy-coded, with different combinations of shared and area-specific parameters. We evaluated if models with area-specific parameters led to better fit and quantified the differences in area-specific parameters (indexed by subscripts heat and ref). The model with all area-specific parameters can be written as:

$$L_i \sim \text{Student-}t\left(v,\ \mu_i, \sigma\right) \tag{1}$$

$$\log(\mu_i) = A_{\text{ref}} \log\left[L_{\infty\text{ref}j[i]}\left(1 - e^{\left(-K_{\text{ref}j[i]}\left(t - t_{0\text{ref}j[i]}\right)\right)}\right)\right] + $$
$$A_{\text{heat}} \log\left[L_{\infty\text{heat}j[i]}\left(1 - e^{\left(-K_{\text{heat}j[i]}\left(t - t_{0\text{heat}j[i]}\right)\right)}\right)\right] \tag{2}$$

$$\begin{bmatrix} L_{\infty\text{ref}j} \\ L_{\infty\text{heat}j} \\ K_{\text{ref}j} \\ K_{\text{heat}j} \end{bmatrix} \sim \text{MVNormal}\left(\begin{bmatrix} \mu_{L_{\infty\text{ref}j}} \\ \mu_{L_{\infty\text{heat}j}} \\ \mu_{K_{\text{ref}j}} \\ \mu_{K_{\text{heat}j}} \end{bmatrix}, \begin{bmatrix} \sigma_{L_{\infty\text{ref}j}} & 0 & 0 & 0 \\ 0 & \sigma_{L_{\infty\text{heat}j}} & 0 & 0 \\ 0 & 0 & \sigma_{K_{\text{ref}j}} & 0 \\ 0 & 0 & 0 & \sigma_{K_{\text{heat}j}} \end{bmatrix}\right) \tag{3}$$

where log lengths are Student-t distributed to account for extreme observations, $v$, $\mu$, and $\sigma$ represent the degrees of freedom, mean, and the scale parameter, respectively. $A_{\text{ref}}$ and $A_{\text{heat}}$ are dummy variables such that $A_{\text{ref}} = 1$ and $A_{\text{heat}} = 0$ if it is the reference area, and vice versa for the heated area. The multivariate normal distribution in **Equation 3** is the prior for the cohort-varying parameters $L_{\infty\text{ref}j}$, $L_{\infty\text{heat}j}$, $K_{\text{ref}j}$, and $K_{\text{heat}j}$ (for cohorts $j = 1981,\ldots,1997$) (note that cohorts extend further back in time than the catch data), with hyper-parameters $\mu_{L_{\infty\text{ref}}}$, $\mu_{L_{\infty\text{heat}}}$, $\mu_{K_{\text{ref}}}$, $\mu_{K_{\text{heat}}}$ describing the population means and a covariance matrix with the between-cohort variation along the diagonal. We did not model a correlation between the parameters, hence off-diagonals are 0. The other seven models include some or all parameters as parameters common for the two areas, e.g., substituting $L_{\infty\text{ref}j}$ and $L_{\infty\text{heat}j}$ with $L_{\infty j}$. To aid the convergence of this non-linear model, we used informative priors chosen after visualizing draws from prior predictive distributions (**Wesner and Pomeranz, 2021**) using probable parameter values (**Figure 2—figure supplement 7**; **Figure 3—figure supplement 3**). We used the same prior distribution for each parameter class for both areas to not introduce any other sources of differences in parameter estimates between areas. We used the following priors for the VBGE model: $\mu_{L_{\infty\text{ref,heat}}} \sim N(45, 20)$, $\mu_{K_{\text{ref,heat}}} \sim N(0.2, 0.1)$, $t_{0\text{ref,heat}} \sim N(-0.5, 1)$, and $v \sim \text{gamma}(2, 0.1)$. $\sigma$ parameters, $\sigma_{L_{\infty\text{ref}}}$, $\sigma_{L_{\infty\text{heat}}}$, $\sigma_{K_{\text{ref}}}$, $\sigma_{K_{\text{heat}}}$ were given a Student-$t(3, 0, 2.5)$ prior.

We also compared how body growth scales with body size (in contrast to length vs age). This is because size-at-age reflects lifetime growth history rather than current growth and may thus be large because growth was fast early in life, not because current growth rates are fast (**Lorenzen,**

*2016*). We therefore fit allometric growth models describing how specific growth rate scales with length: $G = \alpha L^\theta$, where $G$, the annual specific growth between year $t$ and $t+1$, is defined as: $G = 100 \times \left( \log \left( L_{t+1} \right) - \log \left( L_t \right) \right)$ and $L$ is the geometric mean length: $L = \left( L_{t+1} \times L_t \right)^{0.5}$. Here we use back-calculated length-at-age, resulting in multiple observations per individual. As with the VBGE model, we dummy-coded area to compare models with different combinations of common and shared parameters. We assumed growth rates were Student-t distributed, and the full model can be written as:

$$G_i \sim \text{Student-}t \left( v, \ \mu_i, \sigma \right) \tag{4}$$

$$\mu_i = A_{\text{ref}} \left( \alpha_{\text{ref}j[i],k[i]} L^{\theta_{\text{ref}}} \right) + A_{\text{heat}} \left( \alpha_{\text{heat}j[i],k[i]} L^{\theta_{\text{heat}}} \right) \tag{5}$$

$$\alpha_{\text{ref,heat}j} \sim N \left( \mu_{\alpha_{\text{ref,heat}j}}, \ \sigma_{\alpha_{\text{ref,heat}j}} \right) \tag{6}$$

$$\alpha_{\text{ref,heat}k} \sim N \left( \mu_{\alpha_{\text{ref,heat}k}}, \ \sigma_{\alpha_{\text{ref,heat}k}} \right) \tag{7}$$

We assumed only $\alpha$ varied across individuals $j$ within cohorts $k$ and compared two models: one with $\theta$ common for the heated and reference area, and one with an area-specific $\theta$. We used the following priors: $\mu_{\alpha_{\text{ref,heat}}} \sim N \left( 500, \ 100 \right)$, $\theta_{\text{ref,heat}} \sim N \left( -1.2, \ 0.3 \right)$ and $v \sim \text{gamma} \left( 2, 0.1 \right)$. $\sigma$, $\sigma_{\text{id : cohort}}$ and $\sigma_{\text{cohort}}$ were all given a Student-$t$ $\left( 3, 0, 13.3 \right)$ prior.

We estimated total mortality by fitting linear models to the natural log of catch (CPUE) as a function of age (catch curve regression), under the assumption that in a closed population, the exponential decline can be described as $N_t = N_0 e^{-Zt}$, where $N_t$ is the population at time $t$, $N_0$ is the initial population size and $Z$ is the instantaneous mortality rate. This equation can be rewritten as a linear equation: $\log \left( C_t \right) = \log \left( v N_0 \right) - Zt$, where $C_t$ is a catch at age $t$, if a catch is assumed proportional to the number of fish (i.e. $C_t = v N_t$). Hence, the negative of the age slope is the mortality rate, $Z$. To get catch-at-age data, we constructed area-specific age-length keys using the sub-sample of the total (female) catch that was age-determined. Age length keys describe the age proportions of each length category (i.e. a matrix with length category as rows, and ages as columns). The age composition is then estimated for the total catch based on the 'probability' of fish in each length category being a certain age. Due to the smallest and youngest fish not being representatively caught with the gillnet, the catch is dome-shaped over size and age. We therefore followed the practice of selecting only ages on the descending right limb (*Dunn et al., 2002*; *Figure 4—figure supplement 1*). We fit this model with and without an *age × area*-interaction, and the former can be written as:

$$\log \left( CPUE_i \right) \sim \text{Student-}t \left( v, \ \mu_i, \sigma \right) \tag{8}$$

$$\mu_i = \beta_{0j[i]} \, area_{\text{heat}} + \beta_{1j[i]} \, area_{\text{ref}} + \beta_{2j[i]} \, age + \beta_{3j[i]} \, age \times area_{\text{heat}} \tag{9}$$

$$
\begin{bmatrix}
\beta_{0j} \\
\beta_{1j} \\
\beta_{2j} \\
\beta_{3j}
\end{bmatrix}
\sim \text{MVNormal}
\left(
\begin{bmatrix}
\mu_{\beta_{0j}} \\
\mu_{\beta_{1j}} \\
\mu_{\beta_{2j}} \\
\mu_{\beta_{3j}}
\end{bmatrix}
,
\begin{bmatrix}
\sigma_{\beta_{0j}} & \rho\beta_{0j}\beta_{1j} & \rho\beta_{0j}\beta_{2j} & \rho\beta_{0j}\beta_{3j} \\
\rho\beta_{1j}\beta_{0j} & \sigma_{\beta_{1j}} & \rho\beta_{1j}\beta_{2j} & \rho\beta_{1j}\beta_{3j} \\
\rho\beta_{2j}\beta_{0j} & \rho\beta_{2j}\beta_{1j} & \sigma_{\beta_{2j}} & \rho\beta_{2j}\beta_{3j} \\
\rho\beta_{3j}\beta_{0j} & \rho\beta_{3j}\beta_{1j} & \rho\beta_{3j}\beta_{1j} & \sigma_{\beta_{3j}}
\end{bmatrix}
\right) \tag{10}
$$

where $\beta_{0j}$ and $\beta_{1j}$ are the intercepts for the reference and heated areas, respectively, $\beta_{2j}$ is the age slope for the reference area and $\beta_{3j}$ is the difference between the age slope in the reference area and in the heated area. All parameters vary by cohort (for cohort $j = 1981, \ldots, 2000$). We use the default *brms* priors for these models, i.e., flat priors for the regression coefficients (*Bürkner, 2017*) and $v \sim \text{gamma} \left( 2, 0.1 \right)$. $\sigma$ and $\sigma_{\beta_{0j}, \ldots, 3j}$ were given a Student-$t$ $\left( 3, 0, 2.5 \right)$ prior.

Lastly, we quantified differences in the average age and size distributions between the areas. We estimate the biomass size-spectrum exponent $\gamma$ directly, using the likelihood approach for binned data, i.e., the *MLEbin* method in the R package *sizeSpectra* (*Edwards, 2020*; *Edwards et al., 2020*; *Edwards et al., 2017*). This method explicitly accounts for uncertainty in body masses *within* size classes (bins) in the data and has been shown to be less biased than regression-based methods or the likelihood method based on bin midpoints (*Edwards et al., 2020*; *Edwards et al., 2017*). We pooled all years to ensure negative relationships between biomass and size in the size classes (as the sign of the relationship varied between years). We also fitted lognormal models as data are positive and tailed to length- and age-resolved catch data. Here, we assume that the catchability with respect to size does not differ between the areas, and, therefore, use the entire catch

(*Figure 6—figure supplement 1*). In contrast to the catch curve regression, we do not need to filter representatively caught size or age classes. The lognormal models fitted to age or size (denoted $y_{age, length, i}$) model can be written as:

$$y_{age,\ length,i} \sim \text{LogNormal}\left(\mu_i, \sigma\right) \tag{11}$$

$$\mu_i = \beta_{0j[i]}\left(area_{\text{heat}}\right) + \beta_{1j[i]}\left(area_{\text{ref}}\right) \tag{12}$$

$$\begin{bmatrix} \beta_{0j} \\ \beta_{1j} \end{bmatrix} \sim \text{MVNormal}\left(\begin{bmatrix} \mu_{\beta_{0j}} \\ \mu_{\beta_{1j}} \end{bmatrix}, \begin{bmatrix} \sigma_{\beta_{0j}} & \rho\beta_{0j}\beta_{1j} \\ \rho\beta_{2j}\beta_{0j} & \sigma\beta_{1j} \end{bmatrix}\right) \tag{13}$$

where $\beta_{0j}$ is the intercept for the reference area and $\beta_{1j}$ is the intercept for the heated area. These intercepts vary by year (for years $j = 1987, \ldots, 2003$). We use flat priors for the regression coefficients, and $\sigma$ was given a Student-$t$ $(3, 0, 2.5)$ prior, and compared models with and without random slopes.

All analyses were done using R (*R Development Core Team, 2020*) version 4.0.2 with R Studio (2021.09.1). The packages within the *tidyverse* (*Wickham et al., 2019*) collection was used to process and visualize data. Models were fit using the R package *brms* (*Bürkner, 2018*). For the non-linear von Bertalanffy growth equation and the allometric growth model, we used informative priors to facilitate convergence. These were chosen by defining vague priors, and then progressively tightening these until convergence was achieved (*Bürkner, 2017*; *Gesmann and Morris, 2020*). We used prior predictive checks to ensure the priors were suitable (vague enough to include also unlikely predictions, but informative enough to ensure convergence), and the final prior predictive checks are shown in *Figure 2—figure supplement 1* and *Figure 3—figure supplement 1*. We also explored priors vs posteriors to evaluate the influence of our informative priors visually (*Figure 2—figure supplement 7*; *Figure 3—figure supplement 3*). For the linear models (catch curve and mean size), which do not require the same procedure to achieve convergence typically, we used the default priors from *brms* as written above. We used three chains and 4000 iterations in total per chain. Models were compared by evaluating their expected predictive accuracy (expected log pointwise predictive density) using leave-one-out cross-validation (LOO-CV) (*Vehtari et al., 2017*) while ensuring Pareto $k$ values <0.7, in the R package *loo* (*Vehtari et al., 2020*). Results of the model comparison can be found in the *Supplementary file 1*. We used *bayesplot* (*Gabry et al., 2019*) and *tidybayes* (*Kay, 2019*) to process and visualize model diagnostics and posteriors. Model convergence and the fit were assessed by ensuring potential scale reduction factors ($\hat{R}$) were less than 1.1, suggesting all three chains converged to a common distribution (*Gelman et al., 2003*), and by visually inspecting trace plots, residuals QQ-plots, and with posterior predictive checks (*Figures 2–4*, *Figure 6—figure supplement 2*).

## Acknowledgements

We thank all staff involved in data collection, Jens Olsson and Göran Sundblad for discussions, Christine Stawitz and an anonymous reviewer for feedback that greatly improved the manuscript, and Forsmark Kraftgrupp AB for making data publicly available. This study was supported by SLU Quantitative Fish and Fisheries Ecology.

## Additional information

### Funding

No external funding was received for this work.

### Author contributions

Max Lindmark, Conceptualization, Formal analysis, Supervision, Investigation, Visualization, Methodology, Writing – original draft, Writing – review and editing; Malin Karlsson, Formal analysis, Investigation, Writing – original draft; Anna Gårdmark, Data curation, Supervision, Writing – review and editing

## Author ORCIDs

Max Lindmark http://orcid.org/0000-0002-3841-4044
Anna Gårdmark http://orcid.org/0000-0003-1803-0622

## Decision letter and Author response

Decision letter https://doi.org/10.7554/eLife.82996.sa1
Author response https://doi.org/10.7554/eLife.82996.sa2

---

## Additional files

### Supplementary files

• Supplementary file 1. Expected log pointwise density (elpd) for different growth models. (a) Comparison of von Bertalanffy growth models with different combinations of shared and area-specific parameters (ordered by the difference in expected log pointwise density (elpd) from the best model). Note that in all models, $L_{\infty j}$ and $K_j$ vary among cohorts. (b) Comparison of allometric growth models with common or unique $\theta$-parameter (exponent in the allometric growth model), ordered by the difference in expected log pointwise density (elpd) from the best model.

• MDAR checklist

### Data availability

All raw data and R code to clean and reproduce the results reported in this paper are available on GitHub (copy archived at *Lindmark, 2023*) and have been deposited on Zenodo.

The following dataset was generated:

| Author(s) | Year | Dataset title | Dataset URL | Database and Identifier |
|---|---|---|---|---|
| Lindmark M | 2023 | maxlindmark/warm-life-history: v1.0-accepted | https://doi.org/10.5281/zenodo.7853025 | Zenodo, 10.5281/zenodo.7853025 |

---

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
