## [Editor Report]

This work provides convincing evidence to refute a general tenet in biology, that warming induces smaller maximum body sizes in adult ectoterm individuals. Using a semi-natural experiment in an exceptional man-made ecosystem, the authors demostrate that fish in waters warmed by a nearby nuclear plant grew faster but died younger, causing little effect on the size distribution of fish in the area. This work will be of interest to ecologists and physiologists interested in the impacts of global warming on natural communities.

---

## [Decision Letter]

**Decision letter after peer review:**

Thank you for submitting your article "Faster growth rates and higher mortality but similar size-spectrum in heated, large-scale natural experiment" for consideration by *eLife*. Your article has been reviewed by 2 peer reviewers, and the evaluation has been overseen by a Reviewing Editor and Meredith Schuman as the Senior Editor. The following individual involved in the review of your submission has agreed to reveal their identity: Christine Stawitz (Reviewer #1).

Essential revisions:

Both reviewers raised questions about the validity/meaning of CPUE values for different treatments. Please revise in response to these comments and amend the text when possible.

*Reviewer #1 (Recommendations for the authors):*

Congratulations on an excellent paper – I really enjoyed reading it.

58: Is this a common way of measuring changes in size? Looking strictly at mean body size (w/o age structure) seems simplistic – but a citation would convince me this is a common practice

240 – 244: There was a bit more information in your Github repository – it looks like you started with the default priors then had to tighten them to achieve convergence in some of the VBGF models? I agree this is too much detail to include in the main body of the paper but for future scientists' use, I would recommend adding a supplementary section describing how you did this and a table with the priors you used so readers can find them without going to Github.

376: please add the full citation.

Figure 4A: Does it mean anything that CPUE has a lower rate at the beginning of the time series in the heated area than the unheated area? Or is the slope the only thing that matters?

Equations 1- 12: I find it confusing to reuse mu, eta, and σ throughout the three different modeling approaches and would prefer to see a unique letter applied for each of the three models to improve clarity.

*Reviewer #2 (Recommendations for the authors):*

I'd like to see more consideration of faster life history vs higher mortality in this paper. Have these fish adapted a faster life history in response to warmed conditions that happen to result in higher mortality because of the demands of the faster pace of life? Or, given that adults did not shrink, is the increased mortality due to some other factor, like increased predation or stress from the heat? Perhaps this cannot be answered in this system, but it seems like a worthwhile consideration, especially given the unexpected results.

I did have some more substantial questions on the CPUE section, which I'll try to put together here. It seems that CPUE is always lower for the heated population. Could the unexpected increased size observed in heated adults be due to a tradeoff between slower growth due to temperature but higher growth due to a release from density dependence? I think the authors suggest this might be the case, but it would be helpful to state it more explicitly.

Can the authors explain how the initial population size is known and whether it differs between the heated and reference population? It seems there are many reasons why reproductive potential could differ between the heated and reference populations, especially if warming shifts reproductive investment (as would be expected). Why does the CPUE figure start at age 3? It seems younger fish were caught and the fishbase entry on perch says they can mature earlier than 3. Is the mortality rate they measure only applied to fish that have been 'recruited' to the survey? And if fish can mature younger than 3, are some being missed by looking at CPUE for 3+ only?

Finally, the grammar could use some editing, although the errors do not impact readability. I've made a few suggestions below but did not point out all places where grammar could be improved.

---

## [Author Response]

Reviewer #1 (Recommendations for the authors):Congratulations on an excellent paper – I really enjoyed reading it.58: Is this a common way of measuring changes in size? Looking strictly at mean body size (w/o age structure) seems simplistic – but a citation would convince me this is a common practice

Thank you. We think statements/predictions about shrinking (without specific mentioning of the age and/or size-structure) often is made when talking about “large scale” or general shrinking of organisms as a universal rule. The temperature-size rule (TSR) on the other hand, which we describe in more detail further down (line 68) describes changes in size over ontogeny, and hence does not make predictions about the size-distribution of organisms or populations, which is also affected by other processes such as mortality. The reason we open the introduction with the universal prediction about shrinking (and the papers we cite here do focus on average shrinking of species or taxa, to address your specific question) is to make it clear that the predictions about organism shrinking with warming is based on various, sometimes related, biogeographical pattens across space and time, and on the other hand, individual-level experimental rules. Therefore, we think there is a need to distinguish these patterns and rules because there are situations where they do not conform. The example we give is that the TSR predicts larger size-at-age for young specimens, meaning that mortality (and/or shrinking of adults) must increase to offset that size-increase for population-level shrinking to occur.

We have edited the second sentence (lines 64–66) to make it clearer.

240 – 244: There was a bit more information in your Github repository – it looks like you started with the default priors then had to tighten them to achieve convergence in some of the VBGF models? I agree this is too much detail to include in the main body of the paper but for future scientists' use, I would recommend adding a supplementary section describing how you did this and a table with the priors you used so readers can find them without going to Github.

Thank you for pointing out that need. Perhaps a detail, but for the record, *brms* forces the user to specify priors in non-linear models, so we never started with the standard default priors that are used for general regression parameters. Instead, we first defined broad priors centered on what we believed were reasonable values for this fish species, and then progressively tightened them until convergence was achieved. We have added text in the model fitting paragraph (lines 330–338) describing the iterative approach to finding priors for the non-linear model and removed the single sentence describing this approach under each model description and a reference. All priors are given in the main text under each model with in-line equations, so perhaps it is not needed to put them also in a table.

376: please add the full citation.

Done

Figure 4A: Does it mean anything that CPUE has a lower rate at the beginning of the time series in the heated area than the unheated area? Or is the slope the only thing that matters?

We are not sure what is meant by “a lower rate of CPUE (catch per unit effort) in the beginning of the time series”, but we suspect the reviewer means that the catch rate is lower in the heated area on average (since *Figure 4* depicted data and the global prediction without the random cohort effect). For mortality estimates, the slope is what matters, because it relates to the instantaneous mortality parameter in a population with exponential decay when log transformed, whereas the intercept does not matter directly for the mortality estimate. Note also that we have added a model for the average age, which is complementary to the analysis of mortality but is conceptually a different approach to showing the same thing: that the heated population consists of younger fish. As also Reviewer#2 pointed out, we do observe a difference in CPUE between the areas (i.e., the intercept), but we can only speculate why this is (possibly linked to size of the habitat).

Equations 1- 12: I find it confusing to reuse mu, eta, and σ throughout the three different modeling approaches and would prefer to see a unique letter applied for each of the three models to improve clarity.

We thought about this but decided that it would probably be clearer to have a unique letter for the response variable, and a general definition for the parameters of the distribution, e.g., L_i_ for length and G_i_ for growth and mu for the mean of the distribution in question. The alternative would be to add more subscripts, but we have already 3 levels of subscripts in some models, and adding another level make it difficult to read. Instead, we hope the structure of the methods (one paragraph per model) the variable spelled out or abbreviated makes it clear that these are separate models.

Reviewer #2 (Recommendations for the authors):I'd like to see more consideration of faster life history vs higher mortality in this paper. Have these fish adapted a faster life history in response to warmed conditions that happen to result in higher mortality because of the demands of the faster pace of life? Or, given that adults did not shrink, is the increased mortality due to some other factor, like increased predation or stress from the heat? Perhaps this cannot be answered in this system, but it seems like a worthwhile consideration, especially given the unexpected results.

It is difficult to say with the data we have available, and it becomes speculative. Since there are other fish predators (mainly pike and larger perch), which also need to feed at higher rates in the warmer water, it seems inevitable that higher predation mortality to some degree contributes to the higher mortality in perch. However, we do not know nor have any way of testing with the data we have available how important this is compared to e.g., physiological stress, ageing and pace-of-life responses in the heated area experience. We do agree however, that it is an important point, and now bring forth this in the introduction, on lines 112–113, and in the discussion, line 497.

I did have some more substantial questions on the CPUE section, which I'll try to put together here. It seems that CPUE is always lower for the heated population. Could the unexpected increased size observed in heated adults be due to a tradeoff between slower growth due to temperature but higher growth due to a release from density dependence? I think the authors suggest this might be the case, but it would be helpful to state it more explicitly.

Yes, we agree that could be part of the explanation. As we replied to Reviewer#1, the issue is that the effect of living in a smaller area (essentially a small lake compared to the open coast, which could affect density), is largely confounded with the temperature effect. That said, there exists data from a few years prior to the warm water pollution in the heated basin (but after its construction), that indicate the CPUE was always lower in the basin (Huss et al., 2019) (doi: 10.1111/gcb.14637). The lower CPUE could therefore be an artefact of the enclosing of the basin, possibly related to its size. However, in those first years of the enclosing, but before the warm water pollution started, growth rates had not yet differentiated (Huss et al. 2019). However, the effect of density also depends on food availability. While prey data are scarce, Sandström et al. (1995) (doi: 10.1111/j.1095-8649.1995.tb01932.x) show that the density of benthic prey was relatively constant over time (including 4 years prior to warming), suggesting that it is the warming itself that drives the difference in size. We have expanded on this in the discussion (lines 516–518).

Can the authors explain how the initial population size is known and whether it differs between the heated and reference population? It seems there are many reasons why reproductive potential could differ between the heated and reference populations, especially if warming shifts reproductive investment (as would be expected). Why does the CPUE figure start at age 3? It seems younger fish were caught and the fishbase entry on perch says they can mature earlier than 3. Is the mortality rate they measure only applied to fish that have been 'recruited' to the survey? And if fish can mature younger than 3, are some being missed by looking at CPUE for 3+ only?

First, we just want to clarify that the initial size here refers to N_0_, the recruitment or abundance at age 0, not the first population size in time. This is not directly observed but is the intercept in the catch curve regression. It likely differs between the areas for many reasons: e.g., the size of the system (potentially affecting the overall abundance and density of the heated population), and differences in reproductive investment due to life history optimization in warmer waters. However, with the latter it is less known if this also translates to a difference in actual “recruitment” to the fishable population, i.e., the abundance at age 3, due to unknown mortality rates early in life.

It is true that fish younger than age 3 were caught. But age 3 is the age when the catches start to descend, and it is thus assumed that only fish above this age are caught representatively (i.e., that catches of age 2 fish are lower is not because these individuals are rarer in the population, but because they are not caught as effectively in the size-selective gill nets). Therefore, using fish older than 2 years is done to ensure that the catchability of certain ages does not affect the slope estimate, it should only be the true catch rate that does that. Ideally, we would want to avoid filtering the catch data like this, but this is the only way to handle the age/size-based catchability of the fishing gear, and it is standard practice in catch curve regression (Dunn et al., 2002) (doi: https://doi.org/10.1016/S0165-7836(01)00407-6).

We have added a figure illustrating the dome-shaped relationship between CPUE and age that is due to both catchability and declines in abundance by age (*Figure 4—figure supplement 1*), and we describe this procedure in the main text now (lines 280–283 and see also lines 310–312).

Finally, the grammar could use some editing, although the errors do not impact readability. I've made a few suggestions below but did not point out all places where grammar could be improved.

Thank you for pointing that out. We have now checked and corrected errors throughout the manuscript.